

# Selection of reference genes for RT-qPCR studies in blood of beluga whales (*Delphinapterus leucas*)

I-Hua Chen[1], Jiann-Hsiung Wang[1], Shih-Jen Chou[1], Yeong-Huey Wu[2], Tsung-Hsien Li[3], Ming-Yih Leu[3,4], Wen-Been Chang[3] and Wei Cheng Yang[1]

[1] Department of Veterinary Medicine, National Chiayi University, Chiayi, Taiwan, ROC
[2] Department of Veterinary Medicine, National Pingtung University of Science and Technology, Pingtung, Taiwan, ROC
[3] Department of Biology, National Museum of Marine Biology and Aquarium, Pingtung, Taiwan, ROC
[4] Graduate Institute of Marine Biology, National Dong Hwa University, Pingtung, Taiwan, ROC

## ABSTRACT

Reverse transcription quantitative PCR (RT-qPCR) is used for research in gene expression, and it is vital to choose appropriate housekeeping genes (HKGs) as reference genes to obtain correct results. The purpose of this study is to determine stably expressed HKGs in blood of beluga whales (*Delphinapterus leucas*) that can be the appropriate reference genes in relative quantification in gene expression research. Sixty blood samples were taken from four beluga whales. Thirteen candidate HKGs (*ACTB*, *B2M*, *GAPDH*, *HPRT1*, *LDHB*, *PGK1*, *RPL4*, *RPL8*, *RPL18*, *RPS9*, *RPS18*, *TFRC*, *YWHAZ*) were tested using RT-qPCR. The stability values of the HKGs were determined by four different algorithms. Comprehensive analysis of the results revealed that RPL4, PGK1 and ACTB are strongly recommended for use in future RT-qPCR studies in beluga blood samples. This research provides recommendation of reference gene selection, which may contribute to further mRNA relative quantification research in the peripheral blood leukocytes in captive cetaceans. The gene expression assessment of the immune components in blood have the potential to serve as an important approach to evaluating cetacean health influenced by environmental insults.

## INTRODUCTION

Reverse transcription quantitative PCR (RT-qPCR) is considered the ideal method in gene expression studies because of its high sensitivity, time efficiency, and reliability (*Derveaux, Vandesompele & Hellemans, 2010*; *Pfister, Tatabiga & Roser, 2011*). In gene expression analysis using RT-qPCR, different starting amounts of messenger RNA between samples and different efficiencies of reverse transcriptases and polymerases can be adjusted by relative quantification, which uses a reference gene (often the housekeeping gene, HKG) as an internal control to calculate target gene (e.g., cytokine gene) expression levels. HKG is required for the maintenance of basic cellular function, and is expressed in all types of cells (*Pfaffl, 2004*), and its expression level is described as stable. However, *Brinkhof et al.*

Corresponding author
Wei Cheng Yang, jackywc@gmail.com

*(2006)* reported that, in dogs, the most stable control genes were ribosomal protein S5 in the liver, kidney, and mammary glands, beta 2-microglobulin (*B2M*) in the left ventricle, and ribosomal protein L8 (*RPL8*) in the prostate, indicating each tissue type has its specific stably-expressed HKG even within the same species. *Vorachek, Bobe & Hall (2013)* and *Vorachek et al. (2013)* reported that for neutrophils, the most stable gene was glucose-6-phosphate dehydrogenase (*G6PD*) in sheep, while in bovine calves, the most stable genes were phosphoglycerate kinase I (*PGK1*) and tyrosine 3-monooxygenase/tryptophan 5-monooxygenase activation protein zeta (*YWHAZ*); however, *G6PD* was ranked fifth in 10 genes tested. It has been suggested that using an inappropriate reference gene could lead to incorrect normalized data, leading to misinterpretation of the results (*Dheda et al., 2005*). Therefore, selecting a suitable reference gene is needed when studying a new species or tissue type.

Cytokine gene expression research has been conducted in both free-ranging and human-cared cetaceans. Studying the correlation between cytokine gene expression and pollutants in free-ranging cetaceans can make these mammals useful sentinels for indicating the environmental status (*Beineke et al., 2007*; *Buckman et al., 2011*). Cytokine gene expression analysis has also been used as a diagnostic tool in analyzing immune status and stress induced by capture–release assessment in dolphins (*Mancia, Warr & Chapman, 2008*). Moreover, it has been used to evaluate the effectiveness of vaccine treatment and implicate the best duration for vaccination in human-cared cetaceans (*Sitt et al., 2010*). Most of the cetaceans in human care facilities have been trained to undergo voluntary blood collection, and the examination frequency can be increased when intensive monitoring is needed. The quantitative analysis of cytokine gene expression in cetacean blood could offer information, in addition to regular blood examination, for estimating the immune status of the animal and facilitating the medical treatment and health management. The most important first step to obtain an accurate assessment of cytokine gene expression in cetacean blood samples is determining the most stably expressed HKG as the reference gene. The purpose of this study is to select the reference gene in blood samples from beluga whales (*Delphinapterus leucas*), which are one of the most commonly found cetacean species in human care. It would provide fundamental and practical information for the quantitative analysis of cytokine gene expression and contribute to preventive medicine and early diagnosis in human-cared cetaceans.

## MATERIALS AND METHODS

### Sample collection and preservation

The voluntary blood collection of beluga was performed in accordance with international guidelines, and the protocol has been reviewed and approved by Council of Agriculture of Taiwan (Approval number 1020727724). Sixty blood samples from four beluga whales (15 from each one) in National Museum of Marine Biology and Aquarium in Taiwan were taken monthly routine or occasionally assessment from 2011 to 2013. It has been suggested to include samples in different experimental groups or different conditions for reference gene selection (*Dheda et al., 2005*). Samples were from beluga whales with

**Table 1 Function, symbol and name of HKGs in this study.**

| Function | Gene | Name |
|---|---|---|
| Carbohydrate metabolism | GAPDH | Glyceraldehyde-3-phosphate dehydrogenase |
| | PGK1 | Phosphoglycerate kinase 1 |
| | LDHB | Lactate dehydrogenase B |
| Ribosomal protein | RPS9 | Ribosomal protein S9 |
| | RPL4 | Ribosomal protein L4 |
| | RPL8 | Ribosomal protein L8 |
| | RPL18 | Ribosomal protein L18 |
| | RPS18 | Ribosomal protein S18 |
| MHC | B2M | $\beta$-2-microglobin |
| Transporter | TFRC | Transferrin receptor |
| Cytoskeleton | ACTB | $\beta$-actin |
| Signal | YWHAZ | Tyrosine 3-monooxygenase/tryptophan 5-monooxygenase activation protein zeta |
| Others | HPRT1 | Hypoxantine phosphoribosyltransferase 1 |

various body conditions including clinically healthy condition (30 samples from four animals), inflammation (six samples from four animals), skin lesions (nine samples from two animals), and internal diseases with various abnormalities in blood work and cytology (15 samples from three animals). Five hundred microliter EDTA-anticoagulated whole blood was fixed in 1.3 mL RNA*later*® (Ambion, Foster City, CA, USA) within 5 min after drawn. Samples were stored at 4 °C in the first 24 h, and then moved to −20 °C for long-term storage.

## RNA extraction and cDNA synthesis

Total RNA of the samples was extracted using Ribo-Pure™ -Blood kit reagent (Ambion) according to the manufacturer's instructions. RNA Armor™ Reagent (ProTech, Pittsburgh, PA, USA) was added into RNA solution to eliminate contaminated RNase. RNA concentration was determined using Qubit™ fluorometer with Quant-iT™ RNA Assay Kit (Invitrogen, Carlsbad, CA, USA). RNA quantity of all samples was adjusted into 100 ng to keep all the samples on the same starting basis. RNA was treated with genomic DNA (gDNA) wipeout solution (Qiagenen, Valencia, CA, USA) before added into reverse transcription working solution. RNA samples after gDNA elimination were tested using qPCR directly to ensure no residue gDNA, which would interfere the analysis of mRNA expression. QuantiTect® Reverse Transcription kit (Qiagen), provided blend of oligo-dT and random primers, was used for cDNA synthesis. Complementary DNA and the remaining extracted RNA were put into −80 °C for long-term storage.

## Primer and probe design

Sequences of the 13 candidate cetacean HKGs (*ACTB*, *B2M*, *GAPDH*, *HPRT1*, *LDHB*, *PGK1*, *RPL4*, *RPL8*, *RPL18*, *RPS9*, *RPS18*, *TFRC*, *YWHAZ*) were obtained from bottlenose dolphin, striped dolphin, beluga whale, killer whale and fin whale (*Balaenoptera physalus*) from GenBank (Tables 1 and 2). Besides 11 HKGs have been evaluated or used in previous studies (*Beineke et al., 2004*; *Beineke et al., 2007*; *Buckman et al., 2011*; *Mancia, Warr &*

**Table 2** Name, accession number, primer sequence, probe number, amplicon size, efficiency and $R^2$ of 13 candidate HKGs.

| HKG name | Accession number | Primer Sequence (5′ − 3′) | UPL Probe Number | Amplicon Size (bp) | Threshold | Efficiency (%) ± SD | $R^2$ |
|---|---|---|---|---|---|---|---|
| ACTB | AB603937.1 | F-AGGACCTCTATGCCAACACG | 157 | 75 | 0.02 | 97.69 ± 1.15 | 0.999 |
| | | R-CCTTCTGCATCCTGTCAGC | | | | | |
| B2M | DQ404542.1 | F-GGTGGAGCAATCAGACCTGT | 93 | 78 | 0.035 | 95.81 ± 0.61 | 0.999 |
| | | R-GCGTTGGGAGTGAACTCAG | | | | | |
| GAPDH | DQ404538.1 | F-CACCTCAAGATCGTCAGCAA | 119 | 81 | 0.02 | 97.03 ± 1.32 | 1.000 |
| | | R-GCCGAAGTGGTCATGGAT | | | | | |
| HPRT1 | DQ533610.1 | F-GTGGCCCTCTGTGTGCTC | 120 | 81 | 0.012 | 98.17 ± 1.44 | 0.999 |
| | | R-ACTATTTCTGTTCAGTGCTTTGATGT | | | | | |
| LDHB | AB477024.1 | F-TCGGGGGTTAACCAGTGTT | 161 | 78 | 0.005 | 100.49 ± 1.58 | 0.995 |
| | | R-AGGGTGTCTGCACTTTTCTTG | | | | | |
| PGK1 | DQ533611.1 | F-CACTGTGGCCTCTGGCATA | 108 | 84 | 0.015 | 95.47 ± 0.31 | 0.999 |
| | | R-GCAACAGCCTCAGCATACTTC | | | | | |
| RPL4 | DQ404536.1 | F-CAGCGCTGGTCATGTCTAAA | 119 | 108 | 0.035 | 96.91 ± 0.98 | 0.999 |
| | | R-GCAAAACAGCCTCCTTGGT | | | | | |
| RPL8 | GQ141092.1 | F-CCATGAATCCTGTGGAGCAT | 131 | 65 | 0.02 | 101.39 ± 2.47 | 0.997 |
| | | R-GGTAGAGGGTTTGCCGATG | | | | | |
| RPL18 | DQ403041.1 | F-GCAAGATCCTCACCTTCGAC | 93 | 104 | 0.02 | 96.55 ± 0.39 | 1.000 |
| | | R-GAAATGCCTGTACACCTCTCG | | | | | |
| RPS9 | EU638307.1 | F-CTGACGCTGGATGAGAAAGAC | 155 | 77 | 0.02 | 98.96 ± 1.39 | 0.999 |
| | | R-ACCCCGATACGGACGAGT | | | | | |
| RPS18 | DQ404537 | F-GTACGAGGCCAGCACACC | 114 | 90 | 0.02 | 98.46 ± 1.23 | 0.999 |
| | | R-TAACAGACAACGCCCACAAA | | | | | |
| TFRC | DQ533608.1 | F-TCCTTTCCGACATATCTTCTGG | 106 | 73 | 0.02 | 97.79 ± 2.49 | 0.996 |
| | | R-CCGCAGCTTTAAGTGCTCTAGT | | | | | |
| YWHAZ | DQ404539 | F-TCTCTTGCAAAAACGGCATT | 135 | 76 | 0.003 | 98.35 ± 0.66 | 0.992 |
| | | R-TGCTGTCTTTGTATGACTCTTCACT | | | | | |

*Chapman, 2008*; *Martinez-Levasseur et al., 2013*; *Müller et al., 2013*; *Sitt et al., 2008*; *Sitt et al., 2010*; *Spinsanti et al., 2006*; *Spinsanti et al., 2008*), the other 2 genes that could participate in other different cell functions were also included (*Echigoya et al., 2009*; *Kullberg et al., 2006*). Primers and corresponding UPL probes were designed and chosen using Roche UPL design software (ProbeFinder, v.2.49) based on Primer3 software (Table 2). All designed primer pairs were checked by *in silico* PCR algorithm in ProbeFinder, which searches the relevant genome and transcriptome for possible mis-priming sites for either of the PCR primers. Before qPCR experiment, the specificity of primers of 13 candidate genes was confirmed using Fast-Run Hotstart PCR kit (Protech) and electrophoresis.

## Quantitative PCR

Quantitative PCR was conducted on 48-well reaction plates using Eco Real-Time PCR System (Illumina, San Diego, CA, USA). Reactions were prepared in a total volume of 10 µl containing 3 µl 12-fold-diluted cDNA, 0.4 µl of each 10 µM primer, 0.2 µl of
UPL probe (Roche), 5 µl FastStart Essential DNA Probes Master (Roche, Risch-Rotkreuz, Switzerland) and 1 µl of RNase/DNase-free sterile water (ProTech). The thermocycling conditions were set as follows: polymerase activation at 95 °C for 10 min, followed by 45 cycles of denaturation at 95 °C for 10 s, and combined primer annealing/elongation at 60 °C for 30 s. All reactions including no template control (NTC) and plate control were carried out in triplicate. The plate control is a well that carries the same reaction components on every plate, and the quantification cycle (Cq) data from the plate control wells was measuring variation. A consistent Cq value of plate control across plates was obtained allowing the data combination from multiple plates into a single study data set. Baseline value was automatically determined for all plates using Eco Software V4.0. Thresholds for each HKG were determined manually (Table 2). The Cq values in triplicate with standard deviation (SD) <0.5 were averaged as raw Cq value. The five-point (10-fold) standard curve of each probe and primer pair was generated from serial dilution of a nucleic acid template. The PCR amplification efficiency ($E$) and $R^2$ of each probe and primer pair were calculated from the slope of a standard curve using the following equation: $E = (10^{(-1/\text{slope})} - 1) \times 100\%$. The average of at least three $E$ values for each HKG was used as gene-specific $E$ for following relative quantity transformation. This study was conducted according to MIQE (Minimum information for publication of quantitative real-time PCR experiments) guidelines (*Bustin et al., 2009*).

## Data analysis

Corrected Cq values (Cq corr) were transformed from raw Cq values using ΔCq formula, $\text{Cq corr} = \text{Cq}_{min} - \log_2 E^{-\Delta \text{Cq}}$, modified from *Fu et al. (2013)*, where ΔCq is the Cq value of a certain sample minus the Cq value of the sample with the highest expression (lowest Cq, $\text{Cq}_{min}$) of each HKG as calibrator. Stability of all HKGs were evaluated and ranked using algorithms geNorm (*Vandesompele et al., 2002*), NormFinder (*Andersen, Jensen & Ørntoft, 2004*), comparative ΔCt method (*Silver et al., 2006*) and Bestkeeper (*Pfaffl et al., 2004*) based on a web-based analysis tool RefFinder (http://www.leonxie.com/referencegene.php) (*Xie et al., 2011*). RefFinder calculated the geometric mean based on rankings obtained from each algorithm and provides the final comprehensive ranking. Thirty samples were randomly selected from the 60 samples, and the results of HKG ranking using 30 and 60 samples were analyzed comparatively.

## RESULT

$E$ values of the 13 candidate HKGs were between 95.47% and 101.39% that fit the strict acceptable range of 95%–105%, and $R^2$ values were 0.992–1.000 that meet the standard of >0.99 (Table 2). According to the mean Cq value of 60 tested samples, the 13 candidate genes can be divided into two groups: high expression level (Cq < 25) and low expression level (Cq > 25; Fig. 1). *ACTB* showed the highest expression level (Cq = 22.08), while *HPRT1* showed the lowest expression level (Cq = 31.48). All HKGs except *TFRC* displayed a small difference between the maximum and minimum Cq values (<5 cycles). The SD of the Cq value for the plate controls in all experiment was 0.33 (SD < 0.5 is acceptable); therefore, the data of all the plates was combined as one data set.

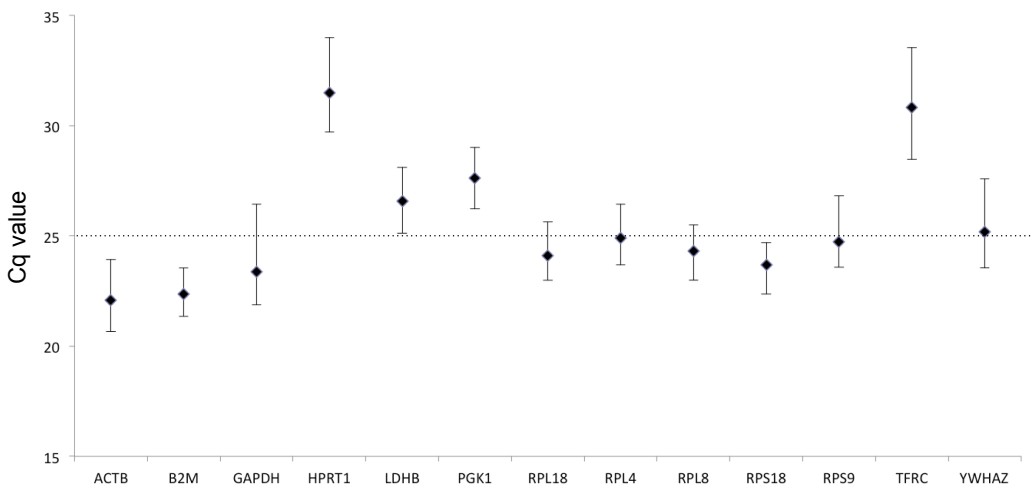

**Figure 1** **Expression levels of candidate HKGs in the tested beluga blood samples ($n = 60$).** Values are given as qPCR cycle threshold numbers (Cq values). Dots represent mean Cq values and whiskers the range of Cq values in the 60 samples.

The commonly used reference gene exploring algorithm, geNorm, calculates the *M* value for gene expression stability based on the geometric mean; a lower *M* value signifies better stability. The gene with highest *M* value (the least stable gene) is excluded, and the highest *M* value gene among the rest of the candidates is continuously excluded to obtain a stability ranking order. *M* values of all the genes were below the default cut-off value ($M = 1.5$), showing good stability for all the genes tested in both 60- and 30-sample groups (Tables 3 and 4). Another value, pairwise variation V, is used to determine the number of reference genes that are required for data analyses. V2/3 values in the 60 and 30 groups were 0.102 and 0.103 (Fig. 2), respectively, which were below the default cut-off value (0.15). It indicated that using two HKGs as reference genes is enough to obtain reliable normalized results in relative quantification. Based on geNorm analysis, *ACTB*, *RPL4*, *PGK1*, and *B2M* were the most stable HKGs in both the 60 and 30 groups (Fig. 3).

The NormFinder program calculates the stability value based on the analysis of gene expression data and ranks the potential reference genes. Lower values are assigned to the most stable genes. The ranking results of NormFinder were essentially identical in both the 60 and 30 groups showing that *PGK1*, *ACTB*, *RPL4*, and *RPL18* were the most stable. The program BestKeeper estimates the expression stability by performing a pairwise correlation analysis of Cq values of each pair of candidate genes. BestKeeper analysis showed that the $SD_{Cq\ value}$ of all HKGs (0.423–0.880) were <1, indicating that these genes were basically stably expressed. The most stable genes identified in the BestKeeper analysis in both the 60 and 30 groups were *RPL8*, *RPS18*, and *B2M*. The comparative $\Delta$Ct analysis is similar to the geNorm program in that the pairs of genes are compared using Cq differences, and those genes are either stably expressed or co-regulated if the $\Delta$Cq values between the pairs of genes remain constant for all samples tested. The best choice in comparative $\Delta$Ct analysis in the 60 and 30 groups was *PGK1*, *RPL4*, and *ACTB*. According to RefFinder, the most
**Table 3** Results of stability among 13 candidate genes computed by four algorithms using 60 beluga blood samples.

| HKGs | Comprehensive ranking | | Delta CT | | BestKeeper | | NormFinder | | geNorm | |
| | Geomean of ranking value | Rank | Average of SD | Rank | SD | Rank | Stability value | Rank | *M* value | Rank |
|---|---|---|---|---|---|---|---|---|---|---|
| RPL4 | 2.3 | 1 | 0.562 | 2 | 0.523 | 7 | 0.319 | 2 | 0.336 | 1 |
| PGK1 | 2.38 | 2 | 0.556 | 1 | 0.595 | 8 | 0.296 | 1 | 0.386 | 4 |
| B2M | 3.08 | 3 | 0.614 | 5 | 0.474 | 3 | 0.418 | 6 | 0.336 | 1 |
| ACTB | 3.57 | 4 | 0.569 | 3 | 0.522 | 6 | 0.326 | 3 | 0.345 | 3 |
| RPL18 | 4.6 | 5 | 0.587 | 4 | 0.509 | 4 | 0.34 | 4 | 0.478 | 7 |
| RPL8 | 4.82 | 6 | 0.664 | 9 | 0.423 | 1 | 0.499 | 10 | 0.46 | 6 |
| RPS18 | 4.86 | 7 | 0.634 | 7 | 0.45 | 2 | 0.466 | 8 | 0.435 | 5 |
| RPS9 | 6.82 | 8 | 0.629 | 6 | 0.712 | 9 | 0.416 | 5 | 0.507 | 8 |
| YWHAZ | 8.43 | 9 | 0.649 | 8 | 0.728 | 10 | 0.454 | 7 | 0.541 | 9 |
| LDHB | 9.64 | 10 | 0.74 | 12 | 0.519 | 5 | 0.594 | 12 | 0.6 | 12 |
| HPRT1 | 10.19 | 11 | 0.674 | 10 | 0.761 | 12 | 0.493 | 9 | 0.564 | 10 |
| GAPDH | 11 | 12 | 0.684 | 11 | 0.759 | 11 | 0.511 | 11 | 0.58 | 11 |
| TFRC | 13 | 13 | 0.956 | 13 | 0.88 | 13 | 0.857 | 13 | 0.655 | 13 |

**Table 4** Results of stability among 13 candidate genes computed by four algorithms using 30 beluga blood samples.

| HKGs | RefFinder | | Delta CT | | BestKeeper | | NormFinder | | geNorm | |
| | Geomean of ranking value | Rank | Average of SD | Rank | SD | Rank | Stability value | Rank | *M* value | Rank |
|---|---|---|---|---|---|---|---|---|---|---|
| PGK1 | 2.21 | 1 | 0.552 | 1 | 0.647 | 8 | 0.26 | 1 | 0.343 | 3 |
| ACTB | 2.45 | 2 | 0.593 | 3 | 0.561 | 6 | 0.356 | 2 | 0.331 | 1 |
| RPL4 | 2.74 | 3 | 0.591 | 2 | 0.564 | 7 | 0.362 | 4 | 0.331 | 1 |
| RPL8 | 4.43 | 4 | 0.678 | 8 | 0.402 | 1 | 0.51 | 8 | 0.432 | 6 |
| RPL18 | 4.53 | 5 | 0.616 | 4 | 0.557 | 5 | 0.359 | 3 | 0.469 | 7 |
| B2M | 4.56 | 6 | 0.637 | 6 | 0.491 | 3 | 0.451 | 6 | 0.364 | 4 |
| RPS18 | 4.7 | 7 | 0.642 | 7 | 0.431 | 2 | 0.473 | 7 | 0.403 | 5 |
| RPS9 | 6.71 | 8 | 0.625 | 5 | 0.788 | 9 | 0.372 | 5 | 0.522 | 9 |
| LDHB | 7.52 | 9 | 0.705 | 10 | 0.497 | 4 | 0.529 | 10 | 0.493 | 8 |
| YWHAZ | 9.72 | 10 | 0.703 | 9 | 0.92 | 11 | 0.513 | 9 | 0.563 | 10 |
| GAPDH | 10.74 | 11 | 0.732 | 11 | 0.87 | 10 | 0.558 | 11 | 0.595 | 11 |
| HPRT1 | 12 | 12 | 0.738 | 12 | 0.951 | 12 | 0.565 | 12 | 0.617 | 12 |
| TFRC | 13 | 13 | 1.023 | 13 | 0.975 | 13 | 0.926 | 13 | 0.68 | 13 |

stable HKGs in the 60 group were *RPL4*, *PGK1*, *B2M*, and *ACTB*, while the most stable HKGs in the 30 group were *PGK1*, *ACTB*, *RPL4*, and *RPL8* (Fig. 3).

## DISCUSSION

The four algorithms used to assess the stability of HKGs, geNorm, NormFinder, BestKeeper, and comparative ΔCt represent feasible strategies, although none of them are currently considered to be the best. BestKeeper uses raw Cq data instead of the relative expression

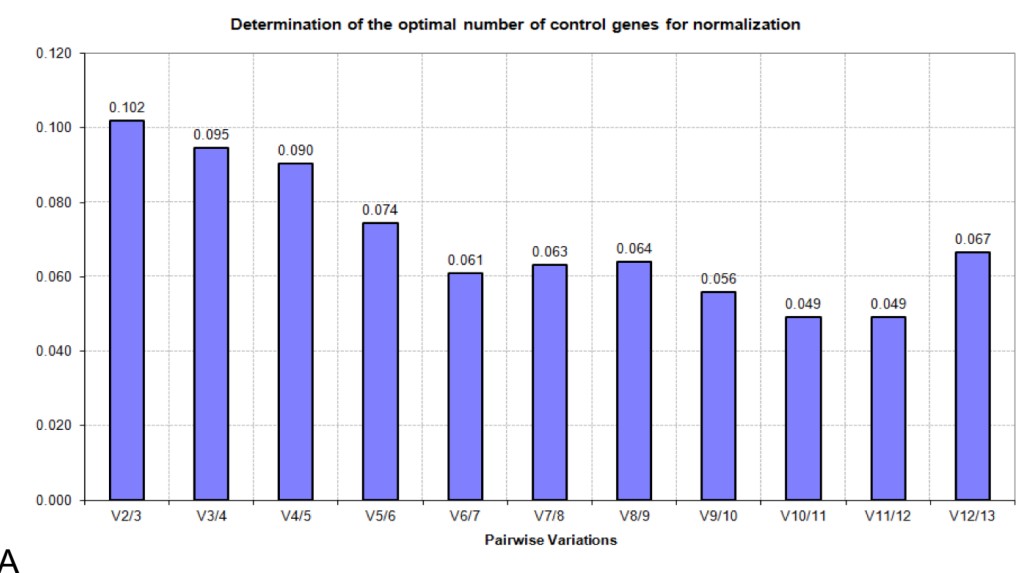

A

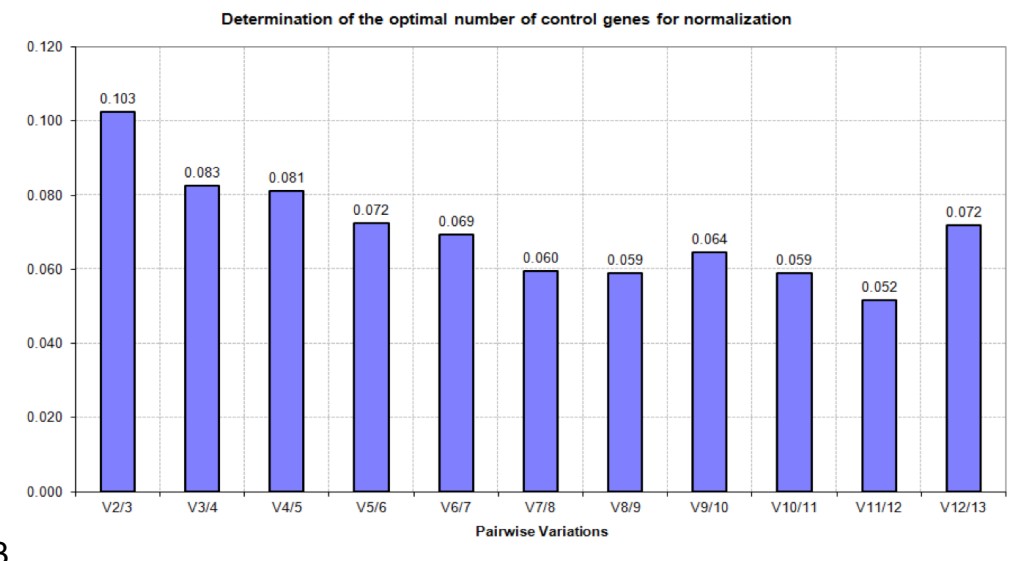

B

**Figure 2** Pairwise variations generated by geNorm algorithm: (A) 60 samples; (B) 30 samples.

level employed by geNorm and NormFinder for selecting the least variable gene, and it has been shown that this may lead to the different outputs among these three methods (*Scharlaken et al., 2008*). Comparative ΔCt and geNorm, which use a pairwise comparison approach, identified the most stable genes by assuming that HKGs are not co-regulated. This may lead to incorrect ranking results when co-regulated genes are included in the analysis (*He et al., 2008*). The NormFinder is likely less affected by co-regulated HKGs because it considers systematic variations through a model-based approach (*Andersen, Jensen & Ørntoft, 2004*). In this study, the HKG stability orders suggested by the four different algorithms were not identical, particularly with the BestKeeper program, which

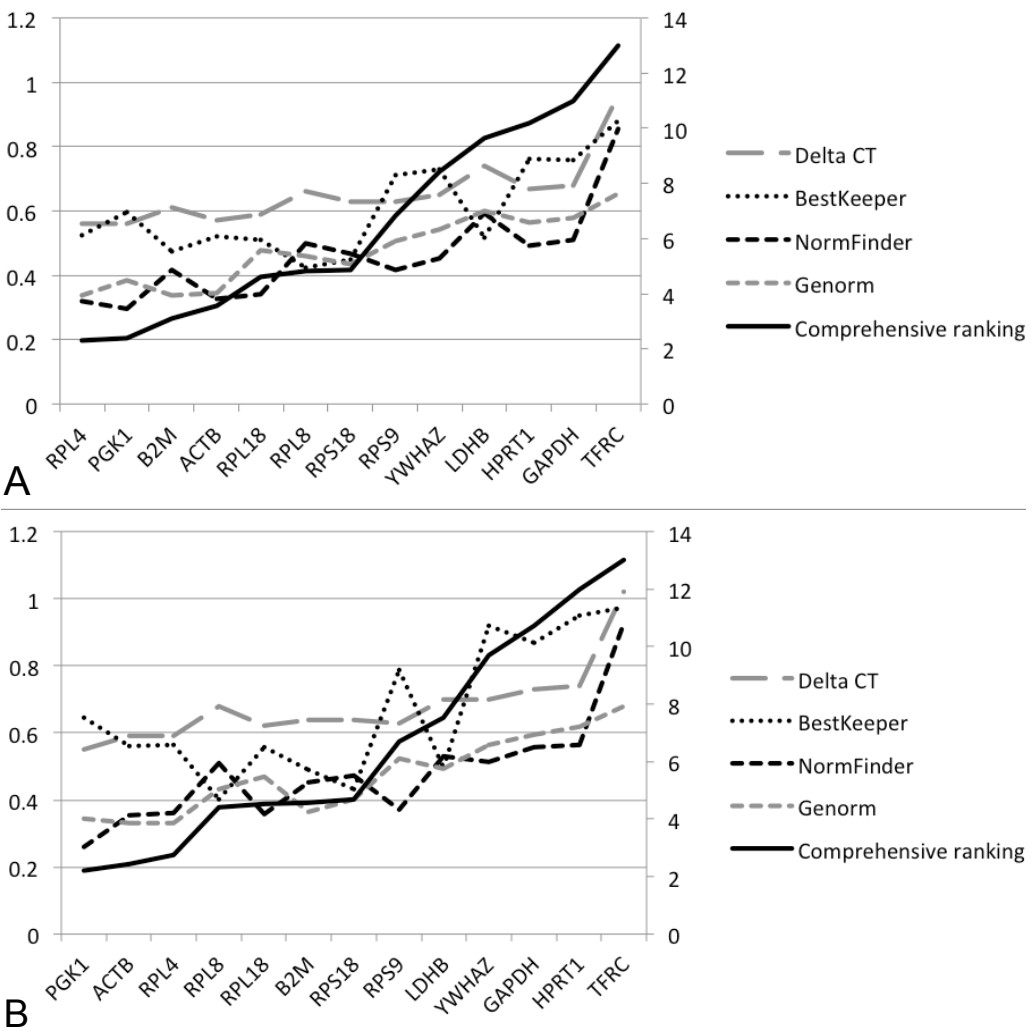

**Figure 3** **Stability values and ranking orders determined by four algorisms and RefFinder: (A) 60 samples; (B) 30 samples.**

could be explained by the distinct principles applied by each of these algorithms. Because these algorithms can demonstrate various rankings of the tested HKGs, in this study RefFinder was used to comprehensively evaluate and rank HKGs based on the rankings from different algorithms.

The four most stable HKGs (*RPL4*, *PGK1*, *B2M*, and *ACTB*) in RefFinder were also in high-ranking orders in NormFinder, geNorm, and comparative $\Delta$Ct, although the ranking in BestKeeper appeared inconsistent with that in the other three algorithms. The $SD_{Cq \, value}$ of these four HKGs (0.474–0.595) showed in the BestKeeper analysis was essentially low indicating these genes were stably expressed. *B2M* encodes for beta-2-microglobulin protein, which is a part of major histocompatibility complex class I molecule. It was shown that a decrease in *B2M* expression is associated with a significant increase in leukocyte counts in dogs (*Piek et al., 2011*), and therefore it might not be an appropriate reference gene for immunology studies. As a result of this report, *RPL4*, *PGK1*, and *ACTB* are strongly
recommended for use in future RT-qPCR studies using beluga blood samples. It has been proposed that the reliability of the normalization factor would increase with the number of stably expressed HKGs included in the calculations (*Vandesompele et al., 2002*). However, in this study the inclusion of more HKGs further reduced the $V$ values. The V2/3 value indicated that it is not needed to include more than two genes into the normalization factor because this would not dramatically improve normalization. Furthermore, it was suggested that one could preferentially choose to use HKGs that have the same expression levels as the target gene in an experimental application to enhance the uniformity of the analysis (*Spinsanti et al., 2006*). According to mean Cq values, *PGK1* was classified in the low expression level group (mean Cq > 25) and the other two genes in the high expression level group (mean Cq < 25). Therefore, it is recommended to use *RPL4* and *PGK1* for low-expression gene studies, such as cytokine expression studies when using beluga blood samples, and *RPL4* and *ACTB* for high-expression gene studies.

In previous studies on reference gene selection in cetaceans, 30 skin biopsy samples in striped dolphins (*Stenella coeruleoalba*) (*Spinsanti et al., 2006*), 20 skin biopsy samples from seven blue whales (*Balaenoptera musculus*), seven fin whales (*Balaenoptera physalus*), and six sperm whales (*Physeter macrocephalus*) (*Martinez-Levasseur et al., 2013*), and 75 blood samples in bottlenose dolphins (*Tursiops truncatus*) (*Chen et al., 2015*) were used. Some practical points, such as available sample sizes and costs of expression stability experiments, may have an effect on the reference gene selection experiments. There is a unique opportunity in this study to compare the HKG expression stability values of 30- and 60-sample groups. The three most stable HKGs were *PGK1*, *ACTB*, and *RPL4* in RefFinder when only 30 randomly selected beluga blood samples were used. The result is consistent with that using 60 samples, only differing in the ranking order of the most stable genes. These three HKGs were the most stable expression genes in geNorm, NormFinder, and comparative $\Delta$Ct, and the $SD_{Cq\ value}$ (0.564–0.647) showed that they were also stably expressed. The result indicated that using only 30 beluga blood samples with various body conditions could select reliable HKGs as reference genes. *Chen et al. (2015)* showed similar results that using 35 bottlenose dolphin blood samples could perform reference gene selection, and *PGK1*, *HPRT1*, and *RPL4* are superior reference genes. *PGK1* and *RPL4* are recommended as reference genes in both beluga whales (in this study) and bottlenose dolphins (*Chen et al., 2015*), and it provides essential information and facilitates future reference gene studies. However, there is still not enough evidence to say that these two genes are the most stable genes in blood samples from toothed whales. Further studies are needed to identify if there are universal reference genes applicable for an accurate normalization of gene expression in cetacean blood samples because of the important value of these animals in various captive environments and the significant susceptibility to environmental degradation in free-ranging species. Cytokine gene expression studies using cetacean blood samples have been conducted using several different HKGs as reference genes, including *GAPDH* and *YWHAZ* in harbor porpoises (*Beineke et al., 2004*; *Beineke et al., 2007*; *Müller et al., 2013*), *GAPDH* in bottlenose dolphins (*Mancia, Warr & Chapman, 2008*), and *RPS9* in bottlenose dolphins, beluga whales, and Pacific white-sided dolphins (*Lagenorhynchus obliquidens*) (*Sitt et al., 2008*; *Sitt et al., 2010*). *RPS9* could potentially be a

suitable reference gene when studying beluga blood samples because in this study it was is ranked in the middle using NormFinder and comparative $\Delta$Ct, and its values in geNorm and BestKeeper were below the default value, indicating basically good expression stability.

We reported the essential background information for the selection of reference genes in RT-qPCR studies of beluga blood samples. A total of 13 candidate HKGs were evaluated, and a suite of best reference genes were recommended to accurately normalize and quantify gene expression in beluga whale blood. To the best of our knowledge, this is the first study to investigate reference gene selection in beluga whales. This investigation is an important basis for future clinical immunology studies in cetaceans.

## ACKNOWLEDGEMENTS

We would like to thank all veterinarians, trainers, students and assistants who participated in this project for their help. We thank two anonymous reviewers for their pertinent and helpful comments.

### Funding

The authors received no funding for this work.

### Competing Interests

The authors declare there are no competing interests.

### Author Contributions

- I-Hua Chen performed the experiments, analyzed the data, wrote the paper, prepared figures and/or tables, reviewed drafts of the paper.
- Jiann-Hsiung Wang and Shih-Jen Chou analyzed the data, contributed reagents/materials/analysis tools, wrote the paper, reviewed drafts of the paper.
- Yeong-Huey Wu performed the experiments, analyzed the data, contributed reagents/materials/analysis tools, wrote the paper, reviewed drafts of the paper.
- Tsung-Hsien Li, Ming-Yih Leu and Wen-Been Chang performed the experiments, analyzed the data, contributed reagents/materials/analysis tools, reviewed drafts of the paper.
- Wei Cheng Yang conceived and designed the experiments, performed the experiments, analyzed the data, wrote the paper, prepared figures and/or tables, reviewed drafts of the paper.

### Animal Ethics

The following information was supplied relating to ethical approvals (i.e., approving body and any reference numbers):

Council of Agriculture of Taiwan (Approval number 1020727724).

### Data Availability

The qPCR raw data can be found in Supplemental Information.

## Supplemental Information

Supplemental information for this article can be found online at http://dx.doi.org/10.7717/peerj.1810#supplemental-information.

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
