# Peer review of "Selection of reference genes for RT-qPCR studies in blood of beluga whales (Delphinapterus leucas)"

_PeerJ, doi:10.7717/peerj.1810_

## Round 0.1 · original submission · Major Revisions

Please address the concerns raised by the reviewers, including validation of the gene specificity for the primers used for qPCR. We hope that you find the comments constructive and look forward to your revised manuscript.

Reviewer 1 ·

Basic reporting

The manuscript is written in good English, easy for understanding and has a clear structure. The purpose of the study was to select the reference genes for RT-qPCR expression analysis of blood samples from beluga whales. Authors provide reliable analysis of expression stability for 13 housekeeping genes in sixty beluga blood samples. The two gene pairs were recommended to use for low expression gene studies (RPL4 and PGK1) and for high expression gene studies (RPL4 and ACTB) as the most stably expressed genes from 13 studied candidates.
Because of authors do not provide comprehensive search of gene candidates (e.g. based on whole transcriptome microarray or sequence data) it should be recommended to provide the basis why (or how) the 13 candidate HKGs were chosen for testing in the study.
It would be helpful for readers to change a line type for DeltaCt and geNorm results on figure 3. It is difficult to identify what is what as it is.
Gene symbols should be marked in italics throughout the text.

Experimental design

The submission clearly defines the research question, which is relevant and meaningful. The investigation has been conducted rigorously and in accordance with MIQE guidelines.

Validity of the findings

In common the data presented in the manuscript are robust and statistically sound. HKG expression stability was tested in 60 samples that seems to be enough for statistics. Authors pointed out that three chosen genes are reliable for clinical immunology studies, however there is no information in the text allowing to assess the representativeness of sample set in this aspect. The information about the number of blood samples collected in inflammation and clinically healthy (and others) body conditions has to be provided.
(Optional) When such information is provided it might be interesting to look at intergroup variation parameter which can be calculated by NormFinder algorithm.

Reviewer 2 ·

Basic reporting

The manuscript “Selection of reference genes for RT-qPCR studies in blood of beluga whales (Delphinapterus leucas)” focus on the selction of endogenous genes to allow an accurate quantification of gene expression in captive beluga whale to evaluate their health status.
The manuscript is an interesting contribution, considering the approach to test four different softwares to select the best reference genes.
However, I have some remarks on the methodological approach outlined in the specific comments in "validity of findings"

Experimental design

no comments

Validity of the findings

Specific comments:
Lines 100-106, Primer and probe design:
How did you check the specificity of the primers? The accession numbers in the table are referred to homologous genes in other cetacean species. It is not clear how you are sure of the complementary of the primers. Did you sequence the genes in beluga? This is not clear from the text and I think it is important for the efficiency of the primer pairs and the whole study.
Line 117: The Cq terms is ambiguous, what does it means? Please use throughout the text Cq or Ct or explain why is Cq used instead of Ct.
Line 124: It is not clear how the calibration curve was done to obtain the efficiency value? How did you calculate it? Did you used a standard curve? How many diluitions?
Discussion, line 215: A similar work was also done on fibroblasts by Spinsanti et al 2008, aquatic toxicology

Additional comments

no comments

---

## Round 0.2 · accepted · Accept

In addition to the evaluation by Reviewer #2, your response to Reviewer #1 was also found to be satisfactory.

Reviewer 2 ·

Basic reporting

No comments

Experimental design

No comments

Validity of the findings

No comments

Additional comments

The authors improved the manuscript properly after the first revision.
Tha paper is now suitable for the publication on Peer J.